# A Retrospective Study on Canine and Feline Mortality during Anaesthesia at a University Clinic in Greece

**DOI:** 10.3390/ani13152486

**Published:** 2023-08-01

**Authors:** Konstantinos Varkoulis, Ioannis Savvas, Tilemachos Anagnostou, George Kazakos, Kiriaki Pavlidou

**Affiliations:** Companion Animal Clinic, School of Veterinary Medicine, Aristotle University of Thessaloniki, 546 27 Thessaloniki, Greece; varkoulisvet@gmail.com (K.V.); tanagnos@vet.auth.gr (T.A.); gkdvm@vet.auth.gr (G.K.); kellypav@gmail.com (K.P.)

**Keywords:** anaesthesia, mortality, dog, cat

## Abstract

**Simple Summary:**

Anaesthesia-related mortality is a critical topic in the field of medicine that should occasionally be reassessed, as it provides information on the quality of anaesthesia care and can be used as an indicator of safety. On top of this, various critical points in the peri-anaesthetic management of the patient can be determined to improve standard of care and avoid future complications. This study investigated cardiac arrest and death rates in dogs and cats during anaesthesia, in a Greek veterinary academic institution and assessed certain factors that may contribute to a higher mortality rate. It was concluded that cats have a higher risk of death than dogs with the current anaesthesia practices, and that more caution must be taken in the peri-anaesthetic management of this species.

**Abstract:**

This retrospective cohort study investigated the mortality rate during anaesthesia and possible contributing factors in canine and feline population in an academic institution in Greece. Data on 1187 dogs and 250 cats which underwent general anaesthesia from 1 January 2018 to 31 December 2019 at the Veterinary Faculty of the Aristotle University of Thessaloniki were analysed regarding cardiac arrest and mortality. In dogs, the rate of cardiac arrest was 1.1% and the rate of death was 0.6%. In cats, these rates were 2.8% and 0.8%, respectively. The mortality rate in healthy/mild disease (ASA I-II) dogs was 0.1% and in cats was 0.5%. Sick (ASA III-V) dogs exhibited a death rate of 2.6%, while sick cats had a rate of 2.2%. In dogs, ASA status had a positive association with cardiac arrest and mortality, with sick dogs being 23 times more likely to suffer cardiac arrest and 24.5 times more likely to die than healthy/mild disease ones. Other factors associated with cardiac arrest and mortality were the anaesthetic protocol and the use of inotropes. In cats, premedication and inotropic support were related to cardiac arrest and death. Feline anaesthesia involves higher risk, and requires greater vigilance in peri-anaesthetic management than dogs.

## 1. Introduction

Anaesthesia-related mortality has been an intensively discussed topic in veterinary and human medicine over the decades, and will remain a major concern for safety and quality of care. Advances in technology and veterinary anaesthesia techniques have led to a more sophisticated approach in peri-anaesthetic monitoring of companion animals. In dogs, anaesthesia-related mortality rates range from 0.009% to 1.29% [1,2,3,4,5,6,7,8,9,10,11,12,13,14,15,16,17,18], whereas in cats the reported mortality rates lie between 0.05% and 2.2% [1,2,3,5,10,11,14,15,17,19,20,21,22].

Nevertheless, in human medicine, anaesthesia-related mortality rates reported among recent studies are far lower. Analysing extracted data from 160 patients out of 217,365 cases of general anaesthesia who suffered cardiac arrest, led to a death rate caused exclusively by anaesthesia of 0.002% [23]. In another study, out of 99 peri-anaesthetic cardiac arrests, there was not a single death caused by anaesthesia [24]. A study with 785,467 patients who underwent general anaesthesia demonstrated a death rate attributed solely to anaesthesia of 0.0005% [25]. Another recent study in Japan including 46,378 patients under general anaesthesia reported that none of 41 peri-anaesthetic deaths were caused exclusively by anaesthesia, though but in ten cases anaesthesia was a contributing factor (0.022%) [26]. These considerably lower mortality rates render obvious the gap between medical and veterinary anaesthesia. In human anaesthesia, various guidelines are frequently and regularly published to establish a uniform and ubiquitous level of high-quality anaesthesia services [27,28]. In veterinary medicine anaesthesia is not strictly provided by specialists, and this fact leads to huge discrepancies in the provided quality of services around the world. It should be noted that in the human literature expert committees revise and analyse patients’ documents and categorise anaesthesia-related deaths according to the causative factor (anaesthesia, procedure, underlying pathology, equipment failure, etc.), distinguishing peri-anaesthetic deaths between those in which anaesthesia was the definite cause and those in which anaesthesia was a contributing factor [25,26,29]. This extensive classification is unfortunately not the case in the veterinary literature, and as a result the anaesthesia-related mortality rates where anaesthesia was the main cause of death may not reflect the true values.

Despite the extended research on this topic, the necessity of further investigation of various peri-anaesthetic factors in order to continuously evaluate the standards of care in clinical practice still exists, as well as the need to strive for state-of-the-art universal guidelines. To the best of our knowledge, a similar study has not been conducted before in Greece. The aim of this study was to assess the mortality rate in dogs and cats during anaesthesia at our institution and explore possible contributing factors. We hypothesized that our mortality rates would be ranged close to the same values reported in recent publications.

## 2. Materials and Methods

This was a retrospective study in dogs and cats that underwent general anaesthesia from 1 January 2018 to 31 December 2019 at the School of Veterinary Medicine, Aristotle University of Thessaloniki, Greece. Mortality was defined as any death that occurred during anaesthesia, i.e., between induction of general anaesthesia with an injectable or volatile anaesthetic agent and cessation of administration of these agents. Exclusion criteria were insufficient data, euthanasia, and more than one incidence of anaesthesia per animal. In cases where the patient underwent more than one anaesthetic event, the most invasive and demanding was recorded, according to ASA status, duration of general anaesthesia and procedure, and type of procedure (surgical interventions were preferred as compared to non-invasive diagnostic procedures).

Data extracted from the anaesthetic records were identification number, date of procedure, species, breed, sex, age, body weight, body condition score, ASA status, premedication, induction and maintenance anaesthetic agents, intubation, inotropic support, analgesics, duration of anaesthesia and surgery, time of cardiac arrest, resuscitation attempts (open or closed), and death.

The age was encoded as “young”, “middle aged”, and “senior”, according to life expectancy and specific characteristics of the patients’ breed [30,31] and size [4,20,32] (Table 1). The main organ system involved was encoded as: “cardiovascular”, “dental”, “digestive”, “ear-nose-throat”, “neoplasia”, “neurological”, “ophthalmic”, “obstetric”, “respiratory”, “skeletal”, “urinary”, and “systemic”. The latter included dysfunction in two or more organ systems. The type of procedure was encoded as: “diagnostic” (minimally invasive procedures, diagnostic imaging, endoscopy), “orthopaedics”, “dental”, “neurosurgery”, “ophthalmic”, and “soft tissue” (including obstetric procedures). The physical status of the animals had been classified by the anaesthetist at the time of anaesthesia administration based on the American Society of Anesthesiologists’ (ASA) classification: ASA I—healthy patient, ASA II—mild systemic disease, ASA III—moderate systemic disease, ASA IV—severe systemic disease that is a constant threat to life, and ASA V—not expected to survive within 24 h without the operation. The same anaesthetist had estimated the body condition score (BCS) of the animals based on the following five categories: 1—severely underweight, 2—underweight, 3—ideal, 4—overweight, and 5—severely obese [33,34].

Premedication drugs were encoded as: “alpha-2 agonists alone”, “alpha-2 agonists with other drugs”, “acepromazine”, “acepromazine with other drugs”, “other drugs”, and “none” for patients that did not receive any premedication. Induction agents were as: “propofol”, “ketamine”, “isoflurane”, or “other drug” (opioids, benzodiazepines, etomidate, etc.). For the maintenance of general anaesthesia, the encoding was: “propofol”, “isoflurane”, “ketamine”, or “opioid with midazolam”. Additional analgesia was encoded as: “NSAID”, “none”, or “other”. In the latter category, steroids and paracetamol were included. Duration of anaesthesia was encoded as “under 30 min”, “between 30 and 120 min”, and “over 120 min”, and duration of surgery was recorded as “under 30 min”, “between 30 and 90 min”, and “over 90 min”.

### Statistical Analysis

Descriptive statistics (mean and standard deviation) were calculated for numerical continuous variables. For the qualitative variable, proportions and odds ratios (OR) were calculated with 95% confidence intervals (CI). The statistical significance level was set to α = 0.05. For all calculations, the statistical software IBM SPSS Statistics v27 was used.

## 3. Results

The overall number of general anaesthesia cases meeting the inclusion criteria was 1555. Twenty-three records with insufficient data were excluded. Cases of animals that underwent anaesthesia more than once were excluded (78 dogs and 17 cats), resulting in a population of 1437 animals with 1187 dogs and 250 cats. After exclusion of euthanised patients, the final population consisted of 1431 animals, with 1182 dogs and 249 cats (Figure 1).

### 3.1. Dogs

Most of the 1182 dogs were mixed-breed (45.5%), followed by Maltese (5.3%), French Bulldog (3.4%), Pit Bull (3.3%), German Shepherd (3.2%), and Yorkshire (3.2%). The sex distribution was 48.6% males and 51.4% females. The age distribution was 47.4% young, 35.3% middle-aged, and 17.3% senior. Most dogs had a body condition score of 3/5 (66.6%). Regarding ASA scale, 434 (36.7%) were assigned to ASA I, 514 (43.5%) to ASA II, 191 (16.2%) to ASA III, 40 (3.4%) to ASA IV, and 3 (0.3%) to ASA V. Thus, most dogs studied (948, 80.2%) were healthy/mild disease animals (ASA I-II), whereas sick dogs (ASA III-V) were fewer in number (234, 19.8%).

Most cases were related to obstetrics, including spays and castrations (21.1%), followed by skeletal (15.9%) and systemic diseases (14.8%), neoplasia (12.2%), and dental (11.1%). For the type of procedure, 47.5% were related to soft tissue surgery, 23.9% to diagnostic, and 14% to dental procedures. Duration of anaesthesia for most cases ranged between 30 and 120 min (50.6%) or over 120 min (42.7%). Duration of surgery for most cases ranged between 30 and 90 min (43.7%), followed by interventions lasting over 90 min (40.8%).

Almost all dogs were intubated (95.9%). The most frequent premedication protocol included alpha-2 agonists exclusively (42.6%) or in combination with other drugs (30.8%), while most animals were induced with propofol (94.6%) and maintained with isoflurane (92.2%). Systemic opioid administration was used in 69.1% of cases, whereas regional anaesthesia was applied in 27.2%; specifically, epidural anaesthesia was utilized in 11.1% of cases. Additional analgesia using non-steroidal anti-inflammatory drugs (NSAIDs) was administered in 48% of dogs, while 49.1% did not receive any additional analgesic drug (NSAID, steroids, paracetamol). Inotropes were used in 7.9% of cases and mechanical ventilation in 6.3%.

Thirteen dogs suffered cardiac arrest (1.1%), out of which seven died (0.6%). More specifically, the rate of cardiac arrest in ASA I-II was 0.2% (2/948 dogs), while the mortality rate of these dogs was 0.1% (1/948 dogs). Dogs of ASA III-V had a cardiac arrest rate of 4.7% (11/234 dogs), while the death rate was 2.6% (6/234 dogs). Of the thirteen dogs that suffered cardiac arrest, six recovered (46.2%). In seven dogs, closed cardiopulmonary resuscitation (CPR) was performed (58.3%) and four of them survived, whereas open CPR was applied in five (41.7%) and two of them survived (Appendix A and Appendix B). One dog that was recorded as having recovered from cardiac arrest without CPR must be a case of incomplete data.

Regarding the thirteen dogs that suffered cardiac arrest, one was ASA I, one ASA II, five ASA III, five ASA IV, and one ASA V. Moreover, one ASA II dog, four ASA III, dogs one ASA IV dog, and one ASA V dog died. There was a positive correlation between ASA grading and cardiac arrest (*p* < 0.0005), as well as between ASA grading and mortality (*p* < 0.0005). Specifically, ASA I-II dogs were associated with lower cardiac arrest and mortality rates, while ASA III-V dogs were associated with higher rates of cardiac arrest and death. Patients were divided into two groups, healthy/mild disease dogs (ASA I-II) and sick ones (ASA III-V). Statistical analysis revealed a significant correlation between these two groups and cardiac arrest (*p* < 0.0005) as well as mortality (*p* < 0.0005). Odds ratio (OR) estimation for risk stratification revealed that ASA III-V dogs were 23 times more likely to suffer cardiac arrest (OR 22.9; CI 95% 5.1–104.2; *p* < 0.0005) and 24.5 times more likely to die compared to ASA I-II dogs (OR 24.5; CI 95% 2.9–204.6; *p* < 0.0005) (Table 2).

There were statistically nonsignificant differences among the age groups, although there was a trend for senior dogs to suffer cardiac arrest more frequently (*p* = 0.077). Regarding anaesthetic drugs, dogs that did not receive premedication or that received drugs other than alpha-2 agonist and acepromazine in their premedication plan were associated with increased rate of cardiac arrest (*p* < 0.0005). Dogs that did not receive premedication were correlated with increased mortality rate (*p* = 0.003). Induction of anaesthesia with drugs other than ketamine, propofol, or isoflurane was associated with higher rates of cardiac arrest (*p* = 0.029) and death (*p* = 0.018). Patients that received a combination of opioids and midazolam for maintenance of general anaesthesia demonstrated higher rates of cardiac arrest (*p* = 0.014) and death (*p* = 0.006). Animals that did not receive additional analgesia were associated with increased rates of cardiac arrest (*p* = 0.014) and death (*p* = 0.033). Patients under inotropic support were twenty times more likely to suffer cardiac arrest (OR 20.2; CI 95% 6.5–63.2; *p* < 0.0005) and nine times more likely to die (OR 8.9; CI 95% 1.9–40.7; *p* = 0.013). Systemic and cardiovascular diseases were correlated with increased cardiac arrest rates (*p* = 0.031). Finally, mechanical ventilation was significantly associated with cardiac arrest (*p* = 0.044).

### 3.2. Cats

The most represented breed of cats was Domestic Short Hair (DSH), with a proportion of 94%. The sex distribution was 51.8% males and 48.2% females. Most cats were young (71.1%) and the rest were middle-aged (18.1%) or senior (10.8%). Most cats had a body condition score of 3/5 (65.5%).For the ASA status in cats, 95 (38.2%) were ASA I, 108 (43.4%) ASA II, 38 (15.3%) ASA III, seven (2.8%) ASA IV, and one cat (0.4%) was ASA V. Consequently, most cats (203, 81.5%) were healthy/mild disease (ASA I-II) and 46 (18.5%) were sick animals (ASA III-V).

Most cases were related to obstetrics (24.9%), followed by skeletal disease (16.1%), dental health problems (15.7%), and systemic disease (14.5%). Regarding the type of procedure, most interventions performed were soft tissue surgeries (45.8%), followed by dental procedures (20.5%) and diagnostics (19.3%). The duration of anaesthesia in most feline cases ranged between 30 and 120 min (58.2%). Most procedures lasted between 30 and 90 min (46.2%).

Most cats under general anaesthesia were intubated (89.2%). Alpha-2 agonists were the main anaesthetic drugs used for premedication, either in combination with other drugs (47%) or as single agents (23.3%). General anaesthesia was induced with propofol (80.7%) and maintained with isoflurane (88.8%). Systemic opioid administration was used in 78.3% of the cases, whereas regional anaesthesia was used in 20.9% of cases. Epidural anaesthesia was performed in 4.4% of cases. Additional analgesia with NSAID was administered in 51% of cats, while 48.6% did not receive NSAID or steroids as adjunctive analgesic. Inotropic support was used in 15.7% of cases, while mechanical ventilation was applied in 2.8% of cases.

Seven cats suffered cardiac arrest (2.8%), out of which two died, resulting in a mortality rate of 0.8%. More specifically, ASA I-II cats had a cardiac arrest rate of 2% (4/203), while the mortality rate was 0.5% (1/203). ASA III-V cats had a cardiac arrest rate of 6.5% (3/46), whereas the death rate was 2.2% (1/46). CPR was performed on all cats, and 5/7 recovered (71.4%). On five of these, closed CPR was selected (71.4%) and four of them survived, while the other two underwent open CPR (28.6%) and only one survived. Out of the seven cats that suffered cardiac arrest, one was ASA I, three ASA II, one ASA III, one ASA IV, and one ASA V. There was a statistically significant association between ASA grading and cardiac arrest (*p* = 0.008). We found statistically nonsignificant difference in cats between ASA status and death (*p* = 0.07), even when the animals were separated into the two groups of healthy/mild disease (ASA I-II) and sick (ASA III-V). Out of those suffering cardiac arrest, one ASA II cat and one ASA IV cat died (Appendix A and Appendix B).

Another factor in cats which was positively correlated with cardiac arrest was BCS, as animals with BCS 2 had increased rates of cardiac arrest (*p* = 0.044). Cats that did not receive any kind of premedication were associated with increased mortality rates (*p* = 0.027), while those that did not receive any premedication or that received sedatives other than alpha-2 agonists and acepromazine had an increased risk of cardiac arrest (*p* = 0.028). Induction of general anaesthesia with isoflurane or ketamine (*p* = 0.007), as well as lack of analgesia (*p* = 0.034), were associated with increased risk of cardiac arrest. Cats under inotropic support demonstrated increased odds of both cardiac arrest (*p* = 0.001) and death (*p* = 0.025) (Table 3).

## 4. Discussion

In our study, the mortality rates in dogs and cats approximated those of some recent studies, but seemed to deviate from others, which is to be expected considering the differences among studies in study design, population, type of procedures, and diversity regarding equipment and personnel in each working environment. It is hard to compare mortality rates among academic institutions/referral clinics and small private practices due to the differences in hospitalized patient populations, with the former encountering more severe and complicated cases (ASA III-V) than the latter ones. Apparently, academic institutions have greater rates of peri-anaesthetic mortality in general than smaller private clinics [1,2,3,4,5,6,10,11,12,13,14,15,16,19,20,35]. A study in the United Kingdom with a large population of patients (98,036 dogs) reported that institutions had a peri-anaesthetic mortality rate of 0.29% compared to private clinics, which had a lower mortality rate of 0.15% [7]. It should be considered that academic institutions serve educational purposes and pre- and post-graduate students may take part in the anaesthetic process, which leaves a greater margin for iatrogenic errors even in the presence of experienced supervisors. In our institution, anaesthesia is provided by interns and residents under the supervision of professors and an ECVAA diplomate. One or two students are assigned to each case as well.

Furthermore, there is disagreement about the definition of anaesthesia-related death. Studies in dogs and cats where death was defined as the negative outcome of cardiac arrest during anaesthesia [3,5] may demonstrate different results in rates than studies in which a clear definition of anaesthesia-related death is provided, excluding surgical causes or underlying pathology related mortality [7,13]. At this point it should be clarified that anaesthesia-related death is a rare event; thus, the qualitative evaluation of safety during the peri-anaesthetic period cannot merely be based on mortality analysis. Therefore, the term anaesthesia-related morbidity must be implemented and gain more attention in future studies. In this way, a broader and more complete safety assessment can take place by including other anaesthesia-related adverse effects such as cardiac arrest, neurologic deficits, etc.

In dogs, our results seem to be in accordance with a recent prospective observational cohort study in 18 veterinary hospitals in Japan with a population of 4310 dogs and an anaesthesia-related mortality rate of 0.65% [13]. The most recent nested case-control study in the United Kingdom, with a population of 157,318 dogs, revealed a mortality rate of 0.1% within the first 48 h and 0.14% within two weeks after the event [18]. It must be noted that in this study dogs that received sedation were included as well, and almost half of the population underwent a routine neuter surgery (89,852 dogs). A retrospective study in the USA with a large population of patients (42,349 dogs) demonstrated an anaesthesia-related death rate in dogs far smaller than ours (0.009% versus 0.6%) [14]. This difference may be attributed to the fact that the focus area of this clinic was neutering, a procedure routinely performed in healthy/mild disease animals (ASA I-II). On the contrary, a retrospective observational study in an American institution reported a death rate of 5.1% [16]. The population of that study was very limited, with only 235 dogs undergoing general anaesthesia, and euthanised animals were included. That study aimed to investigate mortality in the post-operative period. Another matched case-control study looked into anaesthesia records of 1,269,582 dogs across 822 veterinary hospitals in the USA and reported an anaesthesia-related mortality rate of 0.05% [15]. Dogs under sedation were included. The multicentre prospective study of Gil and Redondo, with data across 39 Spanish veterinary clinics and a similar population to that in the current study (2012 dogs) demonstrated an anaesthesia-related mortality rate almost twice as high as ours (1.29%) [12]. A lower death rate (0.18%) was reported in a prospective cohort study by Brodbelt et al., though with a larger animal population (98,036 dogs) [9]. In a retrospective study with a similar animal population (1281 dogs), the anaesthesia-related mortality rate was 0.94% [6], while in one other prospective cohort study with only 942 dogs the death rate was 0.96% [4]. An older prospective study at the University of Colorado reported an anaesthesia-related mortality rate of 0.43% in dogs [2]. Out of 2556 dogs under general anaesthesia, 25 were euthanised and were included in the death rate. Finally, a prospective study from Ontario with 8087 dogs reported an anaesthesia-related mortality rate of 0.11% [3], while an even older prospective study demonstrated a death rate of 0.23% [1]. The level of evidence of the latter studies is low due to inaccurately collected data and statistical analysis performed without computer software.

In cats, our mortality differs from reports in other publications. The reason for this may be the limited population size, with only 249 cats after excluding those euthanised. A recent retrospective study in Florida reported an anaesthesia-related death rate in cats of 0.05%, far lower than ours [14]. Again, this large study with 71,557 cats, exclusively included routine neutering procedures in healthy animals. Another matched case-control study with the largest feline population of the studies cited here (273,684) demonstrated a lower mortality rate of 0.11% (with sedated animals included) [15]. In another prospective cohort study with a large feline population (79,178) the death rate of cats under general anaesthesia was 0.26% [21]. Hosgood and Scholl in their prospective cohort study reported the highest anaesthesia-related mortality rate in the literature at 2.2% [20]. The population size was extremely low, however, with only 138 cats, which decreases the validity of this study. An older prospective study from Colorado with a population of 683 cats reported a death rate of 0.43% (euthanised animals included), almost half the rate that we found [2]. In another prospective study with a population of 8702 cats a mortality rate of 0.1% was reported; however, many of the data were missing in the records [3]. Older studies have demonstrated mortality rates around 0.29% and 0.31% [1,19]. The validity of these studies is low due to inadequate statistical analysis due to lack of computer software; moreover the Dodman et al. study was designed as a questionnaire which was sent to various clinics [19]. Thus, these studies have great limitations and incomplete data collection, which could have led to false results.

Other studies have reported anaesthesia-related mortality rates for both canine and feline species as totals; however, because of differences in population and the definition of mortality these data are incomparable [5,10,11,14,17]. The most recent qualitative observational study, conducted with data on both cats and dogs under general anaesthesia across all Banfield hospitals in the USA, reported a decline in anaesthesia-related mortality from 0.074% in 2017 to 0.062% in 2020 after implementation of certain quality standards [17]. In a recent observational cohort study in France conducted in two stages, the overall anaesthesia-related death rate decreased from 1.35% (a population of 3546 dogs and cats) to 0.8% (a population of 2685 dogs and cats), which is close to our finding (0.63%) [10,11]. Another study conducted in South Africa reported a lower death rate of 0.08% in both species combined [5]. It must be noted that the main type of procedure was neutering and the study design was in the form of a questionnaire, resulting in certain limitations (inaccurate and insufficient data, deviations, etc.) which could lead to falsely lower rates.

In summary, an investigation of the literature reveals no trend in mortality rates. Anaesthesia-related mortality rates in dogs and cats do not decline, increase, or stay fixed over the years. This observation can be justified by the inability to compare studies due to the lack of a similar study designs, and populations, and environments (i.e., institution versus clinic). Nonetheless, a common finding among studies which was evident in ours as well was the fact that cats had higher anaesthesia-related mortality rates than dogs [1,4,7,14,15,20]. While in two studies the rates were similar, none reported higher rates in dogs compared to cats [2,3]. It must be pointed out that the study populations of dogs and cats differed greatly in size, with cats being represented in smaller numbers. Thus, it can be concluded that either cats are misclassified regarding the ASA scale due to insufficient pre-anaesthetic examination or that this species has greater peri-anaesthetic risk compared to dogs with the current veterinary anaesthesia practices.

Our results on cardiac arrest agree with those found in older studies [4,20]. However, two other studies that investigated anaesthesia-related cardiac arrests reported lower rates (0.11–0.43%) [2,3]. In our study, the success rate of CPR was quite high compared to another recent study, where the overall return of spontaneous circulation (ROSC) was 28% but survival to discharge was only 4% [36]. This might be associated with the fact that animals arresting under anaesthesia are easier to resuscitate because the exact time of the arrest is witnessed and the animals are generally well oxygenated before the arrest occurs.

Considering ASA status in dogs, a significant association was demonstrated between cardiac arrest and anaesthesia-related death. This finding is consistently reported in the literature [1,3,4,8,11,12,13,15,18,20]. In cats, ASA status was positively correlated with cardiac arrest but not with mortality, most likely because of the limited feline population size of this study. Most animals that died had a systemic disease as an underlying pathology (ASA III-V). The present study found a statistically non-significant correlation between type of disease and mortality. Only a correlation between systemic and cardiovascular diseases with high risk for cardiac arrest could be demonstrated in dogs. In our study, the cause of death was not further analysed because of insufficient data. For this purpose, more data about the circumstances under which death occurred should be collected, a different study design should be selected, and post-mortem necropsy would be required in order to exclude other causative agents related to equipment and human factors (iatrogenic), intervention or surgery performed, other unknown or undiagnosed underlying pathologies, etc. An older study reported that the cardiovascular and respiratory system accounted for 37% of deaths in dogs and 57% in cats [7]. Another study attributed death in 8/28 animals (28.6%) to tachycardia and hypoxemia from hypoventilation or V/Q mismatch during anaesthesia [13]. Other rare causes of death in that study were multiple organ dysfunction syndrome (17.9%), neurological causes (14.3%), and urinary pathology (14.3%). One of the most recent studies has reported that 69% of anaesthesia-related deaths were attributed to cardiorespiratory factors [18]. It is important to note that all deaths in the present study occurred during maintenance of general anaesthesia (17/20 cardiac arrests). No death was reported during induction of anaesthesia, a period in which the other three arrests took place, all of which were treated successfully. This could be justified by the fact that induction is a short stage which mostly takes place under strict supervision by experienced personnel.

The body condition score was associated with cardiac arrest only in cats, not in dogs. The current study could not demonstrate a correlation between extreme body condition scores with high risk of death, which has been shown in other studies [7,13,15]. It is well known that the closing volume in the lungs of obese patients exceeds the functional residual capacity, which can lead to atelectasis and consequently to hypoxemia, especially during periods of hypoventilation or apnea during induction of general anaesthesia [37,38]. On the contrary, emaciated patients are prone to electrolyte imbalance, hypoproteinemia, and hypothermia, and have depleted physiologic reserves which become noticeable in the presence of perioperative stress. Moreover, overdose of anaesthetics in these patients is common, as adipose tissue plays an important role in the termination of action of various lipophilic drugs in the central nervous system [39].

In our study, age was not associated with cardiac arrest or mortality. This finding is in agreement with the previously cited study from Florida [14]. Nonetheless, plenty of other studies have shown opposite results [4,7,11,12,20]. Age seems to be confounded by ASA status, as shown in an older study, which is something most studies do not take into account [4]. Age is additionally correlated with the breed of the animal, with smaller animals tending to have lower mortality than giant breeds [30].

Furthermore, there was a statistically nonsignificant difference in cardiac arrest and peri-anaesthetic mortality among various breeds in dogs and cats. This could be explained by the fact that certain breeds which require special peri-anaesthetic management, such as Greyhounds, might have been underrepresented in our study population. This could alternatively be the result of an individualised anaesthetic plan adjusted according to the peri-anaesthetic needs of those breeds.

The fact that certain anaesthetic drugs had a statistically nonsignificant difference in cardiac arrest and mortality in certain peri-anaesthetic phases in dogs and cats cannot be easily explained. The same applies for the administration of inotropes in both species. The above-mentioned variables should be analysed under the effect of patients’ ASA status as a confounding factor. It is common practice at our institution to avoid substances such as a2-agonists or acepromazine in ill patients (ASA III-V). Moreover, animals in critical condition (ASA III-V) are more likely to receive inotropic support. The use of epidural and regional anaesthesia did not seem to affect the mortality, which has been reported in another study [10,11]. The same was reported for systemic opioid administration. Nonetheless, lack of additional analgesia with drugs such as NSAIDs, steroids, or paracetamol was associated with higher risk of cardiac arrest and death in dogs. An earlier study reported that dogs receiving opioids and NSAIDs had lower mortality risk compared to patients that did not receive any analgesic [12]. In other words, the statistical significance of these findings should be interpreted with caution because undefined confounding factors might have affected our results.

Intubation nonsignificantly affected cardiac arrest and mortality. The limited number of animals which were not intubated (75 dogs and cats) might have affected this outcome. At our institution, we routinely use lidocaine before intubating cats to avoid the risk of laryngeal spasm [40], which could have averted adverse effects reported in numerous studies [1,3,21].

Type and duration of procedure or anaesthesia non-significantly affected cardiac arrest and mortality. In other studies, demanding type of surgery, urgent interventions, and longer durations of general anaesthesia were associated with increased mortality [7,12,18]. The current study did not evaluate whether procedures were scheduled or urgent; therefore, no comparison was made.

In dogs, mechanical ventilation was associated with higher risk of cardiac arrest. These results might not be representative because of the small number of mechanically ventilated animals (82 dogs and cats). In another study, it was reported that a greater proportion of sick animals ASA IV (35/82) than ASA I (184/614) required mechanical ventilation [6]. Its use necessitates knowledge and experience, as otherwise deleterious outcomes for the patient are to be expected (pneumothorax, decrease in preload, ventilator-induced lung injury, acute respiratory distress syndrome (ARDS), etc.) [41]. Thus, this statistically significant difference could be the result of inadequate adjustment of ventilator settings during the maintenance phase of general anaesthesia.

This retrospective study is accompanied by a number of limitations:The collected data might be incomplete, with certain level of recall bias as well as misclassification bias due to the involvement of many veterinarians in the registration process of ASA status, which can ultimately lead to significant deviations from the true values. With this type of design, it is difficult to determine cause and effect; however, an association among different variables becomes feasible.This type of study requires a large patient population, especially when rare events such as cardiac arrest and anaesthesia-related death are being investigated. The current results are related to our institution only, and do not reflect the overall anaesthesia-related mortality rate of the whole country. The study had a relatively small dog population and an even smaller number of cats enrolled compared to other studies, which decreases the value of the study, especially regarding cats. Animals with incomplete data were excluded from the study; however, none of those suffered cardiac arrest or died. Therefore, the true anaesthesia-related mortality rate would have been lower if those animals had been included in the denominator, especially in cats.Certain parameters such as age, premedication, induction, maintenance, and additional analgesia protocols, as well as inotropic support, should have been analysed in conjunction with ASA status as confounding factors.In this study, mortality was investigated only during anaesthesia; therefore, a significant proportion of postoperative anaesthesia-related deaths must have been obscured, leading to a lower mortality rate. Most studies in the literature emphasize the fact that most anaesthesia-related deaths occur during the postoperative period [6,7,13]. There were cases in which patients were transferred to the ICU intubated and still under general anaesthesia. The outcomes of these patients were not recorded in the collected data.As mentioned above, animals that underwent general anaesthesia more than once were identified, and only the most demanding intervention was recorded. Thus, proportions in case load among the different departments were inevitably changed. Categories such as “diagnostic” included non-interventional cases from various departments, and these had a higher caseload proportion.Anaesthesia-related deaths were not further analysed in this study. Thus, causative agents were not identified, which rendered it impossible to distinguish between those mortality cases in which anaesthesia was the exclusive cause and those in which anaesthesia was a contributing factor or non-contributing factor, as is rationally done in human medicine. This could have led to lower anaesthesia-related mortality rates.The doses of different drugs administered to the patients were not evaluated, which could have influenced the overall outcome in certain cases.

## 5. Conclusions

In our academic institution, the mortality rate in dogs was 0.6%, approximating that reported in two other recent studies [11,13], whereas in cats the death rate was 0.8%. There is great heterogeneity in the animal populations and study designs in the current veterinary literature on this topic; thus, results vary greatly, rendering comparisons among them difficult. Future research on this topic with large populations and a similar study design could serve as an indicator for identification of safety gaps and contribute towards quality improvement, which will hopefully be reflected in decreased anaesthesia-related mortality rates.

## Figures and Tables

**Figure 1 animals-13-02486-f001:**
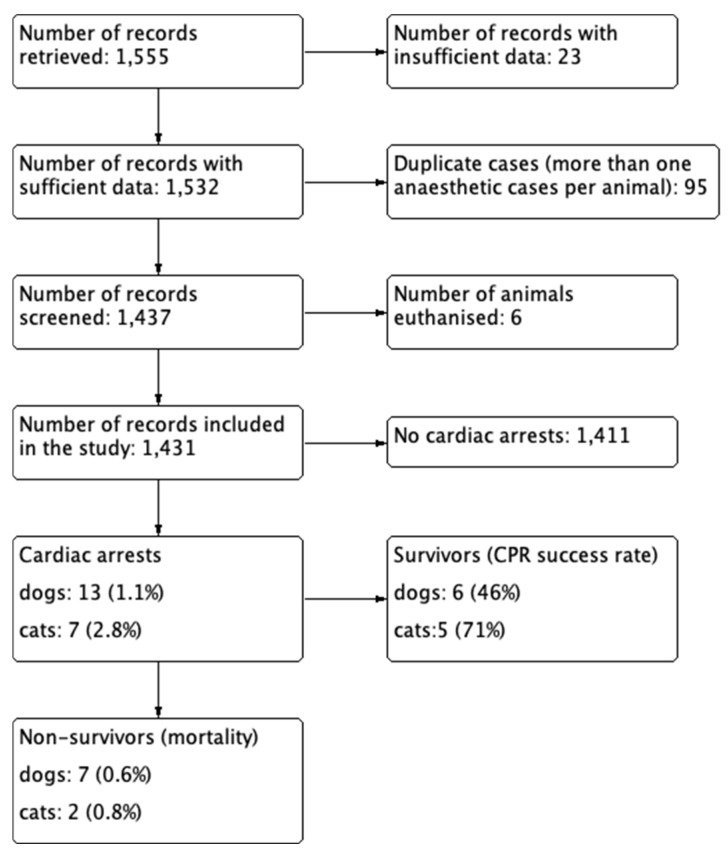
Flowchart of the population selection process.

**Table 1 animals-13-02486-t001:** Age distribution (in years) according to breed size. For certain cases, ranges have been adjusted according to the life expectancy reported for the specific breed [30].

Size	Young	Middle-Aged	Senior
Toy/small dog breeds	<7	7–11	>11
Middle-sized dog breeds	<6	6–10	>10
Big-size dog breeds	<5	5–9	>9
Gigantic-size dog breeds	<3	3–7	>7
Cats	<5	5–9	>9

**Table 2 animals-13-02486-t002:** Factors with statistically significant correlation with cardiac arrest and death in dogs.

Factors	Number of Animals (Total 1187)	Arrest	Death
ASA		*p* < 0.0005	*p* < 0.0005
I	434		
II	515		
III	194 ^a,b^		
IV	41 ^a,b^		
V	3 ^a,b^		
ASA		*p* < 0.0005	*p* < 0.0005
(Ι + ΙΙ)	949		
(III + IV + V)	238 ^a,b^		
Premedication		*p* < 0.0005	*p* = 0.003
alpha-2 agonist	504		
alpha-2 agonist + other	367		
ACP	9		
ACP + other	149		
None	57 ^a,b^		
Other	101 ^a^		
Induction		*p* = 0.029	*p* = 0.018
Isoflurane	2		
Ketamine	9		
Propofol	1123		
Other	53 ^a,b^		
Maintenance		*p* = 0.014	*p* = 0.006
Isoflurane	1095		
Ketamine	1		
Opioid/Midazolam	6 ^a,b^		
Propofol	85		
Additional analgesia		*p* = 0.014	*p* = 0.033
None	585 ^a,b^		
NSAID	567		
Other	35		
Inotropes		*p* < 0.0005	*p* = 0.013
Yes	94 ^a,b^		
No	1093		
Type of disease		*p* = 0.031	*p* = 0.167
Cardiovascular	38 ^a^		
Dental	132		
Digestive	44		
Ear-Nose-Throat	28		
Neoplasia	146		
Neurological	113		
Ophthalmic	19		
Obstetric	249		
Respiratory	31		
Skeletal	188		
Systemic	177 ^a^		
Urinary	22		
Ventilator		*p* = 0.044	*p* = 0.368
Yes	75 ^a^		
No	1112		

^a^: denotes a subset of factor categories with column proportions that differ nonsignificantly from each other at the 0.05 level for cardiac arrest; ^b^: denotes a subset of factor categories with column proportions that differ nonsignificantly from each other at the 0.05 level for death.

**Table 3 animals-13-02486-t003:** Factors with statistically significant correlation with cardiac arrest and death in cats.

Factors	Number of Animals (Total 250)	Arrest	Death
ASA		*p* = 0.008	*p* = 0.073
I	95		
II	108		
III	38		
IV	8		
V	1 ^a^		
ASA			
(Ι + ΙΙ)	203	*p* = 0.125	*p* = 0.341
(III + IV + V)	47		
BCS		*p* = 0.044	*p* = 0.571
1	0		
2	53 ^a^		
3	163		
4	31		
5	3		
Premedication		*p* = 0.028	*p* = 0.027
alpha-2 agonist	58		
alpha-2 agonist + other	117		
ACP	1		
ACP + other	31		
None	9 ^a,b^		
Other	34 ^a^		
Induction		*p* = 0.007	*p* = 0.103
Isoflurane	10 ^a^		
Ketamine	19 ^a^		
Propofol	202		
Other	19		
Additional analgesia		*p* = 0.034	*p* = 0.245
None	122 ^a^		
NSAID	127		
Other	1		
Inotropes		*p* = 0.001	*p* = 0.025
Yes	40 ^a,b^		
No	210		

^a^: denotes a subset of factor categories with column proportions that differ nonsignificantly from each other at the 0.05 level for cardiac arrest; ^b^: denotes a subset of factor categories with column proportions that differ nonsignificantly from each other at the 0.05 level for death.

## Data Availability

Data are available upon request.

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
