# Peer review of "A Retrospective Study on Canine and Feline Mortality during Anaesthesia at a University Clinic in Greece"

_animals, 2023, doi:10.3390/ani13152486_

Round 1

Reviewer 1 Report

Dear colleagues (authors and editors),

It is my pleasure to review the paper entitled: “A retrospective study on canine and feline anaesthetic mortality at a University Clinic in Greece” with the manuscript ID: animals-2498869. Authors address the always interest topic about anaesthetic mortality in small animals. Several previous studies have reported different mortality rates in dogs and cats due to or related with anaesthesia. These rates vary between studies, and probably these differences are mainly because a lack of standardization in the term “anaesthetic mortality” in veterinary science, refinement in small animal anaesthetic techniques, and differences between institutions in which these studies are carried out. In any case, such studies assessing the mortality rate in small animals are always welcome.

This study investigated retrospectively a total of 1.187 dogs and 250 cats which underwent general anaesthesia and were analysed regarding cardiac arrest and mortality. Authors also explored possible contributing factors.  Inclusion and exclusion criteria were clearly stated, and the methodology is simple and sound to my eyes. The results are presented in a clear and structured way. According to authors, mortality rate was 0,6% and 0,8% in dogs and cats, respectively. The conclusions drawn by the authors are in line with the stated objectives.

The present work is a retrospective cohort study, with a relatively small number of animals (some other studies about anaesthetic mortality include several thousands of patients), non-multicentric, in a single academic institution (School of Veterinary Medicine, Aristotle University of Thessaloniki, Greece). Consequently, readers should keep in mind the nature of this research. Along discussion section, authors identify and justify in detail several of the limitations of this study in a very pertinent and appropriate way.

Although this work does not deal with a strictly original topic, after a thorough reading, I conclude that conceptualization, methodology and writing are appropriate, and make this manuscript eligible for publication. Nevertheless, there are some suggestions that I would like to share with authors before publication:

1.     In lines 75-77, authors defined anaesthesia-related death as any death occurred during anaesthesia. It supposes that deaths clearly attributable to surgical procedure (e.g., fatal intraoperative haemorrhage) were included in this study. I would suggest identifying these cases (if any, and if it possible based on anaesthesia records), consider it as an exclusion criterion and remove these cases from the analysis.

2.     I would consider define the following acronyms the firs time they appeared in the manuscript: “NSAID” in line 108, “ROSC” in figure 1, “CPR” in line 161, and “ARDS” in line 421.

3.     In lines 109-111, authors encoded the duration of anaesthesia and the duration of surgery. It seen a bit arbitrary. Do you have any substantial criteria to make such classifications? Could you support it with any reference/s?

4.     In line 123, could you explain the rational to exclude cases of animals that underwent anaesthesia more than once?

5.     Authors divided animals into two groups: “healthy” patients (ASA I and II) and in “sick” patients (ASA III-V). ASA II is a patient with mild systemic disease. So, the term “healthy” sound a bit vague for me to include animals with ASA I or II. I suggest changing the term “healthy” for “healthy or with mild diseases only, without substantive functional limitations” (Portier K and Ida KK (2018) The ASA Physical Status Classification: What Is the Evidence for Recommending Its Use in Veterinary Anesthesia? — A Systematic Review. Front. Vet. Sci. 5:204. doi: 10.3389/fvets.2018.00204).

6.     In results section (lines 172-175, and lines 187-188), authors present odds ratio (OR) data using an inaccurate language as “almost 23 times”, “approximately 20 times”, or “almost 9 times”. I would strongly recommend to present data relative to OR numerically, adding CI95% and p value) (e.g., OR 22.83; CI95% 18.35-25.26; p<0.002).

7.     In line 386, the expression “certain rare breeds” doesn't sound very scientific to me. I would recommend remove the adjective “rare”.

8.     In conclusions section, I would recommend adding the numerical data of anaesthesia-related mortality rate in dogs and cats obtained by authors.

9.     I finally suggest some minor details like:

-        In line 81, please, change “State” by “state”.

-        In line 200, please change “.” by “,”.

-        In line 429, please change “However” by “however”.

Sincerely,

Author Response

The content in the attachment.

Response to Reviewer 1 Comments

Dear colleagues (authors and editors),

It is my pleasure to review the paper entitled: “A retrospective study on canine and feline anaesthetic mortality at a University Clinic in Greece” with the manuscript ID: animals-2498869. Authors address the always interest topic about anaesthetic mortality in small animals. Several previous studies have reported different mortality rates in dogs and cats due to or related with anaesthesia. These rates vary between studies, and probably these differences are mainly because a lack of standardization in the term “anaesthetic mortality” in veterinary science, refinement in small animal anaesthetic techniques, and differences between institutions in which these studies are carried out. In any case, such studies assessing the mortality rate in small animals are always welcome.

This study investigated retrospectively a total of 1.187 dogs and 250 cats which underwent general anaesthesia and were analysed regarding cardiac arrest and mortality. Authors also explored possible contributing factors.  Inclusion and exclusion criteria were clearly stated, and the methodology is simple and sound to my eyes. The results are presented in a clear and structured way. According to authors, mortality rate was 0,6% and 0,8% in dogs and cats, respectively. The conclusions drawn by the authors are in line with the stated objectives.

The present work is a retrospective cohort study, with a relatively small number of animals (some other studies about anaesthetic mortality include several thousands of patients), non-multicentric, in a single academic institution (School of Veterinary Medicine, Aristotle University of Thessaloniki, Greece). Consequently, readers should keep in mind the nature of this research. Along discussion section, authors identify and justify in detail several of the limitations of this study in a very pertinent and appropriate way.

Although this work does not deal with a strictly original topic, after a thorough reading, I conclude that conceptualization, methodology and writing are appropriate, and make this manuscript eligible for publication. Nevertheless, there are some suggestions that I would like to share with authors before publication:

Response: Thank you so much for critically reviewing our manuscript.

Point 1: In lines 75-77, authors defined anaesthesia-related death as any death occurred during anaesthesia. It supposes that deaths clearly attributable to surgical procedure (e.g., fatal intraoperative haemorrhage) were included in this study. I would suggest identifying these cases (if any, and if it possible based on anaesthesia records), consider it as an exclusion criterion and remove these cases from the analysis.

Response 1: Because of the retrospective nature of our study, it was impossible to identify non-anaesthesia related deaths. We have mentioned this as a limitation of our study and changed the phrasing from “anaesthesia-related mortality” to “mortality” throughout the manuscript.

Point 2: I would consider define the following acronyms the firs time they appeared in the manuscript: “NSAID” in line 108, “ROSC” in figure 1, “CPR” in line 161, and “ARDS” in line 421.

Response 2: Corrected.

Point 3: In lines 109-111, authors encoded the duration of anaesthesia and the duration of surgery. It seen a bit arbitrary. Do you have any substantial criteria to make such classifications? Could you support it with any reference/s?

Response 3: This encoding/classification of the duration of anaesthesia was based on Brodbelt, D.C. The Confidential Enquiry into Perioperative Small Animal Fatalities. R. Veterinary Coll. 2006 and Gil, L.; Redondo, J.I. Canine Anaesthetic Death in Spain: A Multicentre Prospective Cohort Study of 2012 Cases. Vet. Anaesth. Analg. 2013, 40, 57–67, which have published similar classification time periods to ours.

Point 4: In line 123, could you explain the rational to exclude cases of animals that underwent anaesthesia more than once?

Response 4: In a cohort study, either retro- or pro-spective, it is methodologically incorrect to include repeated measurements. If for example a patient had been anaesthetise three times, there is a correlation of these three incidents, which affects the variable analysis.

Point 5: Authors divided animals into two groups: “healthy” patients (ASA I and II) and in “sick” patients (ASA III-V). ASA II is a patient with mild systemic disease. So, the term “healthy” sound a bit vague for me to include animals with ASA I or II. I suggest changing the term “healthy” for “healthy or with mild diseases only, without substantive functional limitations” (Portier K and Ida KK (2018) The ASA Physical Status Classification: What Is the Evidence for Recommending Its Use in Veterinary Anesthesia? — A Systematic Review. Front. Vet. Sci. 5:204. doi: 10.3389/fvets.2018.00204).

Response 5: The term “healthy” has been changed to “healthy/mild disease” in ASA I and II patients, throughout text.

Point 6: In results section (lines 172-175, and lines 187-188), authors present odds ratio (OR) data using an inaccurate language as “almost 23 times”, “approximately 20 times”, or “almost 9 times”. I would strongly recommend to present data relative to OR numerically, adding CI95% and p value) (e.g., OR 22.83; CI95% 18.35-25.26; p<0.002).

Response 6: Exact numbers of odds ratios have been included. Please note that odds ratios can only be computed for 2x2 tables.

Point 7: In line 386, the expression “certain rare breeds” doesn't sound very scientific to me. I would recommend remove the adjective “rare”.

Response 7: Corrected.

Point 8: In conclusions section, I would recommend adding the numerical data of anaesthesia-related mortality rate in dogs and cats obtained by authors.

Response 8: Numerical data was added.

Point 9: I finally suggest some minor details like:

-In line 81, please, change “State” by “state”.

-In line 200, please change “.” by “,”.

-In line 429, please change “However” by “however”.

Response 9: All corrected.

Reviewer 2 Report

This study is titled anaesthetic mortality but the methods describe that any animal that went into cardiac arrest during anaesthesia was included whether anaesthesia had anything to do with the arrest or not. Several of the cases suggest that the animals had significant morbidity and that they may have died from their disease rather than from something related to the anesthetic. I think the title should be changed to ‘A retrospective study of cardiac arrest during canine and feline anaesthesia at a …’. This issue should be reflected throughout the paper because the authors have made no attempt to assign the cause of the cardiac arrest to an adverse anesthetic event or to the disease or to a surgical mistake.

Line 38 – ‘The introduction of modern drugs into clinical practice limited significantly major anaesthesia-related complications. Is there a reference for this – what is meant by ‘modern’ – is thiopental included under this definition? Do we have any statistics that show decreased mortality when one compares older drugs to ‘modern’ drugs?

Line 44 – 

Line 54 – ‘Moreover, in veterinary medicine, anaesthesia is not strictly provided by specialists and this fact combined with the often-unbearable financial burden, leads to huge discrepancies in the provided quality of services round the world.’  ‘Moreover’ is a throwaway word and is not needed for this sentence (and not usually used in scientific articles). The statement about specialists and unbearable financial burden does not make sense as written. Do you mean that it would be an unbearable financial burden for anesthesiologists to carry out all anesthetics? I would imagine that even in your hospital the boarded anesthesiologist does not do many cases but supervises others instead. As far as I am aware this would be illegal in most human hospitals in the EU.

Line 76 any death that occurred…

Line 87 – you state ‘The disease state’ and you classify one of these as ‘reproductive’. In the results you refer to spays and castrations as ‘obstetric’ – please use the same terms. Do animals undergoing routine OVH or castration have a ‘disease’ or should their disease state be recorded as ‘healthy’.

Line 96 You state that the ASA status was ‘estimated’ but in line 81 you state that the ASA status was ‘extracted’. Does this mean you revised the ASA status based on a review of the case material? Likewise, for BCS – you extracted the BCI but did you then revise the data. In both of these cases if you revised the recorded data please state how you arrived at the new classification.

Line 122 inclusion rather than including.

Line 123 Cases – were excluded – is this strictly true – as I understand it 78 dogs and 17 cats had two anesthetics (none of them had more than 2?) and you included the anesthetic that represented the ‘most demanding one’. So the case was not excluded but the anesthetic event was? Also wouldn’t it be better to just exclude the first anesthetic since, by definition, the animal had to survive to get to the second anesthetic?

Line 128 – According to your figure, the final population was this number because of the number of animals that were euthanized, not because of the information you describe in the previous sentence.

Figure 1 – this is not just the selection process – you are also presenting results. You start with 1555 cases, but you don’t state how many records were excluded because of insufficient data. You need to standardize the way you are writing the numbers. In most scientific journals decimals are used for values less than 1 and commas are used in values over 1000. E.g. 1,187 dogs and 1.1% in dogs. Please standardize this throughout the manuscript.

Line 150 – ‘while at most’ – while most animal were induced with…?

Line 163 – Of the 13 dogs that suffered…

Line 177 – Here you describe that there was a difference between dogs receiving alpha-2 agonists/acepromazine vs other or no premedication but it is not clear in the table that this is the comparison. Please make it clear what the comparisons are in the table to arrive at the p value.

Line 183 – Patients that received…

Table 3 – left justify your titles as in Table 2.

Discussion

You have not really addressed the nature of the studies. For example, Brodbelt et al. was a prospective study and they did try to separate anesthesia related mortality from procedure/disease related death. Please make sure you talk about the difference between retrospective vs prospective studies. I think it is also worth pointing out that referral practices tend to receive a greater proportion of ASA III-V patients and so the morbidity rate may be higher despite the presence of skilled anaesthetists. So, comparing studies from multiple practices vs a single institution may be different simply because of the different caseload. I think you should also describe how anesthesia is supervised and carried out at your institution. How many of these cases were done by students/interns/residents/boarded anaesthetists?

Line 272 – what is considered a ‘large animal’?

Line 325 – you are making the assumption that these data are comparable – the different study populations and definitions of mortality make this questionable.

Line 339 – this is a  result that should be reported in that section.

Line 390 This paragraph should be rewritten to reflect that the statistical significance of premedication, other induction drugs, inotropic and ventilatory support and minimal use of analgesics are all likely due to the animal being more compromised but you didn’t have enough data to remove the confounding variables.

Line 424 – This paragraph rambles – please try to identify the limitations of your study more succinctly.

The language is unnecessarily flowery and could be edited to be a lot more succinct

Author Response

The content in the attachment:

Response to Reviewer 2 Comments

This study is titled anaesthetic mortality but the methods describe that any animal that went into cardiac arrest during anaesthesia was included whether anaesthesia had anything to do with the arrest or not. Several of the cases suggest that the animals had significant morbidity and that they may have died from their disease rather than from something related to the anesthetic. I think the title should be changed to ‘A retrospective study of cardiac arrest during canine and feline anaesthesia at a …’. This issue should be reflected throughout the paper because the authors have made no attempt to assign the cause of the cardiac arrest to an adverse anesthetic event or to the disease or to a surgical mistake.

Response: Thank you for critically reviewing our manuscript. The title has been changes according to your suggestion.

Line 38 – ‘The introduction of modern drugs into clinical practice limited significantly major anaesthesia-related complications. Is there a reference for this – what is meant by ‘modern’ – is thiopental included under this definition? Do we have any statistics that show decreased mortality when one compares older drugs to ‘modern’ drugs?

Response: The statement about modern drugs has been removed.

Line 44

There was no comment here. Please let us know if we missed something.

Line 54 – ‘Moreover, in veterinary medicine, anaesthesia is not strictly provided by specialists and this fact combined with the often-unbearable financial burden, leads to huge discrepancies in the provided quality of services round the world.’  ‘Moreover’ is a throwaway word and is not needed for this sentence (and not usually used in scientific articles). The statement about specialists and unbearable financial burden does not make sense as written. Do you mean that it would be an unbearable financial burden for anesthesiologists to carry out all anesthetics? I would imagine that even in your hospital the boarded anesthesiologist does not do many cases but supervises others instead. As far as I am aware this would be illegal in most human hospitals in the EU.

Response: We have rephrased this line. Thank you for spotting this.

Line 76 any death that occurred…

Response: Corrected.

Line 87 – you state ‘The disease state’ and you classify one of these as ‘reproductive’. In the results you refer to spays and castrations as ‘obstetric’ – please use the same terms. Do animals undergoing routine OVH or castration have a ‘disease’ or should their disease state be recorded as ‘healthy’.

Response: “Reproductive” has been changed into “obstetric”. All castrations and OVH were ASA I patients.

Line 96 You state that the ASA status was ‘estimated’ but in line 81 you state that the ASA status was ‘extracted’. Does this mean you revised the ASA status based on a review of the case material? Likewise, for BCS – you extracted the BCI but did you then revise the data. In both of these cases if you revised the recorded data please state how you arrived at the new classification.

Response: Corrected and clarified in the manuscript (line 108).

Line 122 inclusion rather than including.

Response: Corrected.

Line 123 Cases – were excluded – is this strictly true – as I understand it 78 dogs and 17 cats had two anesthetics (none of them had more than 2?) and you included the anesthetic that represented the ‘most demanding one’. So the case was not excluded but the anesthetic event was? Also wouldn’t it be better to just exclude the first anesthetic since, by definition, the animal had to survive to get to the second anesthetic?

Response: In a cohort study, either retro- or pro-spective, it is methodologically incorrect to include repeated measurements. If for example a patient had been anaesthetise three times, there is a correlation of these three incidents, which affects the variable analysis. Because of the nature of our study (mortality) we decided to include only the anaesthesia with the higher ASA grading.

Line 128 – According to your figure, the final population was this number because of the number of animals that were euthanized, not because of the information you describe in the previous sentence.

Response: Corrected.

Figure 1 – this is not just the selection process – you are also presenting results. You start with 1555 cases, but you don’t state how many records were excluded because of insufficient data. You need to standardize the way you are writing the numbers. In most scientific journals decimals are used for values less than 1 and commas are used in values over 1000. E.g. 1,187 dogs and 1.1% in dogs. Please standardize this throughout the manuscript.

Response: Corrected.

Line 150 – ‘while at most’ – while most animal were induced with…?

Response: Corrected.

Line 163 – Of the 13 dogs that suffered…

Response: Corrected.

Line 177 – Here you describe that there was a difference between dogs receiving alpha-2 agonists/acepromazine vs other or no premedication but it is not clear in the table that this is the comparison. Please make it clear what the comparisons are in the table to arrive at the p value.

Response: Tables were corrected, and the statistically significant differences are now clear.

Line 183 – Patients that received…

Response: Corrected.

Table 3 – left justify your titles as in Table 2.

Response: Corrected.

Discussion

You have not really addressed the nature of the studies. For example, Brodbelt et al. was a prospective study and they did try to separate anesthesia related mortality from procedure/disease related death. Please make sure you talk about the difference between retrospective vs prospective studies. I think it is also worth pointing out that referral practices tend to receive a greater proportion of ASA III-V patients and so the morbidity rate may be higher despite the presence of skilled anaesthetists. So, comparing studies from multiple practices vs a single institution may be different simply because of the different caseload. I think you should also describe how anesthesia is supervised and carried out at your institution. How many of these cases were done by students/interns/residents/boarded anaesthetists?

Response: Thank you for this comment. We have included some comments in “Discussion”, first paragraph.

Line 272 – what is considered a ‘large animal’?

Response: Corrected and the population size of that study added.

Line 325 – you are making the assumption that these data are comparable – the different study populations and definitions of mortality make this questionable.

Line 339 – this is a result that should be reported in that section.

Response: Added.

Line 390 This paragraph should be rewritten to reflect that the statistical significance of premedication, other induction drugs, inotropic and ventilatory support and minimal use of analgesics are all likely due to the animal being more compromised but you didn’t have enough data to remove the confounding variables.

Response: We clarified that these results are confounded by other factors and thus should be interpreted with caution.

Line 424 – This paragraph rambles – please try to identify the limitations of your study more succinctly.

Response: Rephrased.

Round 2

Reviewer 2 Report

Line 81 & 98 – There is still some confusion about BCI and BCS – please make sure that this is consistent.

Line 87 – I would still argue that animals that were healthy and undergoing routine desexing surgery do not have a ‘disease state’. Please make this something like ‘The main organ system involved was encoded as:’

Line 120 – Please add the records that had insufficient data here – 1555 minus 95 does not equal 1437

Table 2 and 3. I don’t understand the use of the letters. For example in dogs – overall there was a statistical correlation between ASA status and arrest/death. According to your lettering animals in ASA 3-5 had a nonsignificant proportion of animals suffering from arrest/death. If it was not significant, why are you reporting it and why do you need two letters? Perhaps adding two more columns with the ratios of arrest and death here might make this clearer?

Line 220 – Could you please state which of these was successful – did both of the open chest animals survive?

Line 227 – positively correlated

Line 258 – I think this would be a good place to discuss retrospective and prospective studies. The latter generally have a predefined set of criteria for death due to anesthesia. Please make sure that you state whether the studies you are discussing are prospective or retrospective.

Somewhere in this discussion you should address the success of your resuscitation. If you look at the literature on the success of resuscitation the rates are very low. For instance in Dazio et al. (JSAP 2022;64;270-279) the overall return of spontaneous circulation (ROSC) was 28% but survival to discharge only 4%. They noted greater odds of ROSC in patients under anesthesia but not at the level in your study. I have long maintained that animals arresting under anesthesia are easier to resuscitate because the exact time of the arrest is witnessed, and the animals are generally well oxygenated before the arrest occurs – your data further support this idea.

Line 452 – I still think this doesn’t make sense and it should be the last anesthetic that should be included.

The writing is verbose and flowery and needs to be edited extensively to make this paper easier to read

Author Response

The content in the attachment:

Response to Reviewer 2 Comments

Line 81 & 98 – There is still some confusion about BCI and BCS – please make sure that this is consistent.

Corrected.

Line 87 – I would still argue that animals that were healthy and undergoing routine desexing surgery do not have a ‘disease state’. Please make this something like ‘The main organ system involved was encoded as:’

Corrected.

Line 120 – Please add the records that had insufficient data here – 1555 minus 95 does not equal 1437

Excluded records added.

Table 2 and 3. I don’t understand the use of the letters. For example in dogs – overall there was a statistical correlation between ASA status and arrest/death. According to your lettering animals in ASA 3-5 had a nonsignificant proportion of animals suffering from arrest/death. If it was not significant, why are you reporting it and why do you need two letters? Perhaps adding two more columns with the ratios of arrest and death here might make this clearer?

The letters give some more information on which proportion diverse from the expected counts. Groups with same letters do not seem to differ from each other. This is how the statistical package presents the results., e.g. in dogs, ASA 3-5 animals differed significantly from ASA 1-2, that’s why they do NOT have the same letter (a). Groups with SAME letter differ NON-significantly.

If the editor agrees, we can remove the superscripts. Adding more columns would render the tables complicated.

Line 220 – Could you please state which of these was successful – did both of the open chest animals survive?

Information added.

Line 227 – positively correlated

Corrected.

Line 258 – I think this would be a good place to discuss retrospective and prospective studies. The latter generally have a predefined set of criteria for death due to anaesthesia. Please make sure that you state whether the studies you are discussing are prospective or retrospective.

Somewhere in this discussion you should address the success of your resuscitation. If you look at the literature on the success of resuscitation the rates are very low. For instance in Dazio et al. (JSAP 2022;64;270-279) the overall return of spontaneous circulation (ROSC) was 28% but survival to discharge only 4%. They noted greater odds of ROSC in patients under anesthesia but not at the level in your study. I have long maintained that animals arresting under anaesthesia are easier to resuscitate because the exact time of the arrest is witnessed, and the animals are generally well oxygenated before the arrest occurs – your data further support this idea.

Thank you for your valuable comment. Our manuscript has been amended accordingly.

Line 452 – I still think this doesn’t make sense and it should be the last anesthetic that should be included.

We believe that the reviewer might be right, but at this point we cannot re-run the whole analysis for these cases. After all, this would have changed nothing at all in our main results, because no deaths had been recorded in these cases.

We could delete this paragraph if the editor agrees.